# The Value of Fetal Head Station as a Delivery Mode Predictor in Primiparous Women at Term before the Onset of Labor

**DOI:** 10.3390/jcm11123274

**Published:** 2022-06-08

**Authors:** Laurențiu Mihai Dîră, Monica-Laura Cara, Roxana Cristina Drăgușin, Rodica Daniela Nagy, Dominic Gabriel Iliescu

**Affiliations:** 1Doctoral School, University of Medicine and Pharmacy of Craiova, 200349 Craiova, Romania; laurentiu.dira@yahoo.com (L.M.D.); rodica.nagy25@gmail.com (R.D.N.); 2Department of Obstetrics and Gynecology, University Emergency County Hospital Craiova, 200642 Craiova, Romania; roxana.dragusin@umfcv.ro (R.C.D.); dominic.iliescu@umfcv.ro (D.G.I.); 3Ginecho Clinic, Medgin, 200333 Craiova, Romania; 4Department of Public Health, University of Medicine and Pharmacy of Craiova, 200349 Craiova, Romania; 5Department of Obstetrics and Gynecology, University of Medicine and Pharmacy of Craiova, 200349 Craiova, Romania

**Keywords:** fetal head station, digital vaginal examination, delivery mode, vaginal delivery, caesarean section

## Abstract

Objective: Our objective was to demonstrate the role of the clinical determination of fetal head station (FHS) at term to predict the delivery mode in primiparous women before the onset of labor. Methods: This prospective study included unselected primiparous women at term who presented at our tertiary maternity. We excluded multiparous patients, pregnancies with a planned Cesarean section, non-cephalic presentations, and multiple pregnancies. The protocol included weekly clinical examinations to assess the FHS. The results were used to describe the clinical fetal head descent at term. We correlated the fetal head station determinations at each week with labor outcome, including the evaluations performed within the week before delivery. Results: The data show no significant differences between vaginal (VD) and Cesarean section delivery (CS) cases regarding FHS determined at each week at term. The median determinations at the gestational ages (GW) from 37 to 41 were −2 and −3, similar between the two groups, with a more consistent difference at 41 GW: station -1 for VD compared to −3 for CS. There were significant differences between the “week before delivery” evaluations of the two groups. The determinations showed for both groups similar minimum (−5), maximum (+1), and median (−2) FHS values. Most vaginal deliveries cases presented at weekly examinations with increasing rates toward more advanced stations: from 10% at station −4 to 35% at station −1. Although we investigated a low-risk group, we found significant differences between the vaginal and Cesarean groups in terms of age, weight, and BMI. We provided a multiple logistic regression equation that considered the predictive clinical variables at term: the fetal head situation, age, weight, height, and BMI. Conclusion: The clinical evaluation of fetal head station in primiparous before labor onset has a limited value regarding the prediction of the delivery mode. There is a potential benefit for the determinations performed within the week before delivery, but such a policy would require weekly assessments of the FHS at term, which is unlikely to be implemented. Another potential benefit would involve estimating labor outcomes in late-term or prolonged pregnancy. The fine tuning of the logistic prediction should be achieved by increasing the studied population and the number of centers involved before counseling primiparous women at term based on the clinical fetal engagement data.

## 1. Introduction

Fetal head station (FHS) represents the level of the fetal head in the maternal birth canal evaluated by digital vaginal evaluation. According to the American College of Obstetricians and Gynecologists, the pelvic canal of laboring women is divided into 11 levels, also called stations (from −5 to 5), each of them equivalent in centimeters. At station 0, the lowest part of the fetal presentation is at the level of the maternal ischial spines, and this clinical determination is also called fetal head engagement [1]. Higher or lower head stations are expressed in centimeters above (negative) or below (positive) the reference plane. The head engagement can also be evaluated by an abdominal examination. When two-fifths of the fetal head is palpable, the lowest part of the fetal presentation has reached the plane of the maternal ischial spines, corresponding to the descent of the biparietal plane of the fetal head to a level below the pelvic inlet [1,2]. 

Digital vaginal examination (DVE) represents the standard evaluation of fetal head station and position in term pregnancy [3]. Every midwife, resident, or specialist should have proper training and expertise in abdominal and vaginal clinical examination [4,5,6]. Still, the clinical evaluation of fetal head situation appears less accurate than ultrasound determinations [7,8,9] and has low reproducibility [10,11]. The presence of caput succedaneum, a significant molding, or a persistent posterior occipital position [12,13,14] can lead to difficult interpretations [15], but such clinical situations do not occur before the onset of labor. On the other hand, the accuracy of fetal head station assessment before labor is limited because the presenting part is more difficult to palpate through a closed cervix [1].

It is essential to properly assess FHS to accurately evaluate labor progress. In addition, in nulliparous women at term, before labor onset, a high FHS or non-engagement of the fetal head at term is considered a risk factor for CS [16,17,18,19]. The ability to predict the chance of a normal vaginal or operative delivery before the onset of labor would be of great usefulness in daily clinical practice to prevent fetomaternal labor complications while avoiding an increased rate of Cesarean sections, and so far the literature has not proven that it is possible to achieve this [20] However, it is worth continuously trying to predict labor outcome before its onset. Recently, an interesting study by Youssef et al. communicated the impact of dynamic changes of fetal head station on labor outcome [21].

From these premises, in the present study, we aimed to investigate the situation of the fetal head in the primiparous term and whether the clinical evaluation of FHS has a predictive role in estimating the delivery mode before the onset of labor. Therefore, we designed a longitudinal study with weekly determinations starting from 37 GW with several purposes: first, to determine if the evaluation at early term (37 weeks) is predictive for labor outcome; second, to determine a potential descent trend of the fetal head at term; and third, to evaluate the role of the clinical assessments performed within the week before delivery (WBD). 

## 2. Method

We present the prospectively collected data in the Prenatal Diagnostic Unit of the University Emergency County Hospital of Craiova, regional tertiary maternity. We included consecutive low-risk unselected primiparous women admitted for routine 3rd trimester scan, with a gestation age of more than 37 gestational weeks (GW), based on the first trimester dating scan. We excluded cases with indications for elective Cesarean delivery, and high obstetrical risk, e.g., non-cephalic presentation, multiple pregnancies, preeclampsia, diabetes, fetal growth restriction and macrosomia, teenage and elderly primiparous, and prior Cesarean delivery. We did not exclude the cases with induced labor, regardless of the induction outcome. Each patient was adequately informed regarding the study protocol and had to sign an informed consent form to participate. The protocol included weekly digital vaginal examinations of all enrolled patients until delivery. According to each examination, a clinical fetal head station was noted in the database. The clinical examinations were performed by experienced obstetricians (four senior physicians, with more than 5 years of practice). The experience of the physicians was considered as a valuable tool to ensure the objectivity of FHS examinations. There were no cases which reported examination difficulties. Finally, after we documented all the results in the database, we performed a statistical analysis to investigate the predictive value of clinical fetal head station determination on the delivery mode estimation in primiparous women before the onset of labor. The following parameters were considered: FHS at 37GW, the longitudinal variation of FHS during the weekly evaluations, and the situation of the fetal head in the WBD. The Cesarean delivery cases with other than prolonged or arrested labor indication (fetal distress without prolonged labor, placental abruption, etc.) were also excluded from the final analysis.

The trial was approved by the University of Medicine and Pharmacy of Craiova Committee of Ethics and Academic and Scientific Deontology (No.: 18/26.02.2016).

## 3. Statistical Analysis

Descriptive statistics were produced for all study variables (maternal age, height, weight, parity, gestational age, mode and type of delivery, and fetal head station). The data were checked for normality and equal variance using the Kolmogorov–Smirnov test together with the Shapiro–Wilk test. All the continuous variables are presented as the mean and SD or median, if appropriate. The categorical data are presented as frequency and percentage. The results between groups (vaginal delivery and Cesarean delivery) were compared using the Student’s t-test or Mann–Whitney test where applicable with a statistical significance level set at *p* < 0.05. The statistical analysis was performed using IBM SPSS Statistics for Windows, V.22.0. (IBM Corp.).

## 4. Results

### 4.1. Group Characteristics

A total of 276 Caucasian primiparous women from five counties assigned to the University Emergency County Hospital of Craiova were included in the final analysis. Sixteen patients were excluded because they did not comply with the protocol, or the birth information could not be retrieved (5.48% lost to follow-up rate). A total of 74% (N = 204) of the primiparous women were at their first pregnancy. Figure 1 presents the flow chart summarizing the study group. Maternal and labor characteristics are detailed in Table 1.

### 4.2. The Fetal Head Situation at Term

The boxplot representation of the clinical FHS at 37 to 41 GW is presented in Figure 2. The data show no significant differences between vaginal and Cesarean delivery cases in terms of fetal head station determined at each week at term (*p* > 0.05).

The median determinations of clinical FHS at 37 to 41 GW were similar between the two groups: −2.5 vs. −3 at 37 GW, −3 vs. −3 at 38GW, −2 vs. −2 at 39 GW, and −2 vs. −2.5 at 40 GW. A more important difference was noted at 41 GW: station -1 for vaginal deliveries compared with station −3 for Cesarean deliveries.

In the following, we present a detailed analysis of each week at term, underlying the characteristics and differences between the two groups: vaginal and Cesarean section delivery (Figure 3 and Figure 4).

At 37 weeks of gestation, almost one-third (30.6%, n = 41) of the primiparous women who eventually delivered vaginally were diagnosed with a fetal head station of -3, and almost half (48.5%, n = 65) presented with a clinical fetal head station of −2 and −1. In the CS group, 70% (n = 28) presented a higher than −3 FHS (Figure 3 and Figure 4).

At 38 GW, we noted similar percentages of patients in the two groups for all stations, with minor differences of a maximum of 6% (Figure 3 and Figure 4).

At 39 GW, we noted higher rates of CS at high FHS: −5 (10.3% vs. 0.8%), −4 (27.6% vs. 14%), and −2 (31% vs. 27.1%), but surprisingly also at more favorable (lower) clinical head stations: 0 (3.4 % vs. 1.6%) and +1 (6.9% vs. 1.6%). Meanwhile, in the group of vaginal births, we noted higher rates at stations −3 (25.6 vs. 3.4%) and −1 (29.5 vs. 17.2%). The distribution does not respect a certain trend.

A similar variable distribution, where the level of the head station was not correlated with a certain type of delivery, was also noted at 40 weeks of gestation (Figure 3 and Figure 4).

At 41 weeks of gestation, 58.3% (n = 14) of primiparous women who delivered vaginally presented a clinical head station at −1 (vs. 33.3%, n = 1 of the CS), while 66.7% (n = 2) of the primiparous women who delivered by Cesarean section presented a high (−3) pelvic head station, vs. 20.8% (n = 5) of the VDs group (Figure 3 and Figure 4). The differences are visible and important, but they cannot be considered significant because of the low number of pregnancies evaluated.

### 4.3. Week before Delivery

The design of our study enabled us to observe the distribution of the FHS measurements during the WBD in both groups—vaginal and Cesarean deliveries. The clinical evaluations were performed weekly; therefore, the last measurements before labor onset were assigned to “week before delivery” determinations.

The analysis of these determinations showed for both groups similar minimum (−5), maximum (+1), and median (−2) values for the clinical established FHS (Figure 5).

In the group of primiparous women who delivered by Cesarean section, one fifth presented with a clinical fetal head station at −3 (19.6%, n = 11) and −1 (19.6%, n = 11), and one-quarter of cases were noted at the −4 (25%, n = 14) and –2 (26.8%, n = 15) stations (Figure 6). Most of the women that delivered vaginally were found with similar head stations, but with increasing rates toward more advanced stations: from 10% (n = 22) at station −4 to 35% (n = 77) at station −1. The low number of pregnancies that reached stations 0 and +1 do not allow intergroup comparison. Unlike the weekly evaluations, there were significant differences between the WBD evaluations of the two groups (*p* < 0.05).

### 4.4. Longitudinal Evaluation of Fetal Head Station Depending on the Labor Outcome

Given the study’s longitudinal design, consisting of weekly evaluations at term, we aimed to investigate the evolution of the FHS in the two groups (vaginal and Cesarean delivery) during the term weeks. Therefore, we analyzed the incidence of each FHS throughout the weekly evaluation at term, from 37 to 41 GW in vaginal and Cesarean delivery cases (Figure 7 and Figure 8). In both groups, we noted low rates of the extremes, namely stations 0, +1, and −5. In the VD group, the predominant FHS registered were −1, −2, and −3, while the −4 station presented visibly lower rates. Station −4 was better represented in the CS group, in line with the stations mentioned above.

In VD cases (Figure 7), we noted the predominance of stations −3, −2, and −1 with low extremes (stations −5, −4, 0, +1) and a generally similar trend for all weeks. In the CS group (Figure 8), we noted station −4 (red line) at higher rates during weeks 39 and 40, and a generally stationary trend at term.

### 4.5. The Importance of Maternal Clinical Characteristics at Term

We were interested in seeing whether there are significant differences between the two delivery modes regarding the following features: mother’s age, weight, height, and BMI. We have applied the *t*-test for independent variables. The results are shown in Table 2.

From Table 2, we can see that indeed there are significant differences (*p*-value < 0.05) between the two groups in terms of age, weight, and BMI.

Next, we were interested in performing a multiple logistic regression in which the predictive variables considered were the stage of fetal head, age, weight, height, and BMI. The obtained regression equation was:$${\mathrm{birth}}_{\mathrm{mode}}=-0.65-0.23\cdot {\mathrm{fetal}}_{\mathrm{head}}-0.17\cdot \mathrm{age}-0.14\cdot \mathrm{weight}+0.113\cdot \mathrm{height}+0.330\cdot \mathrm{BMI}$$

We also performed an overall goodness of fit through ANOVA. The obtained results in terms of sums of square (SS), degrees of freedom (df), mean squares (MS), F-value, and *p*-level are shown in Table 3.

From Table 3, we can see that indeed these parameters can be considered when predicting the delivery mode (*p*-level < 0.05).

## 5. Discussion

Using the clinical evaluation of the FHS in primiparous women at term before the labor onset, we expected to find a slow downward progression of the fetal head towards station 0, where fetal head engagement occurs, especially in the group of vaginal deliveries. However, our data showed no significant differences between vaginal and Cesarean delivery cases in terms of the FHS determined at each week at term and a stationary situation of the head descent, with a median station of −3 to −2. The only consistent difference was noted at 41 GW: a −1 median station for the VD group compared to −3 for the CS group, but not a statistically significant difference due to the small number of pregnancies reaching this gestational age. These findings suggest that VD is not associated with a better accommodation or engagement of the fetal head in the maternal pelvis at term.

A policy of early-term evaluation of primipara to estimate labor outcome does not appear helpful. There are some differences regarding the head situation at 37 GW: almost half of the patients that eventually delivered vaginally presented with a clinical FHS of −2 and −1, while in the CS group, 70% presented a higher than −3 clinical FHS. However, the statistical analysis yields no significant differences. Moreover, at 38GW, similar rates of patients in the two groups were found for all stations.

In addition, at 39 and 40 GW, we noted a variable distribution that does not respect a specific trend in more advanced term pregnancies, and FHS was not correlated with a specific type of delivery.

At 41 GW, as mentioned above concerning the median station, the differences were more visible: more than half of the primiparous women who delivered vaginally presented a clinical FHS −1 vs. one-third of the CS women. In contrast, two-thirds of the primiparous women that delivered by Cesarean section presented a high (−3) pelvic head station, vs. one-fifth of the VD cases. Apparently, at 41 GW, there is a substantial discrepancy between the groups, and the fetal head is better accommodated with the maternal pelvis in pregnancies finalized with vaginal delivery. Still, the low number of evaluated pregnancies does not allow us to safely consider this difference statisticaly significant. Nevertheless, if further studies confirm our findings, a policy to predict labor outcome in late-term or prolonged pregnancy could be implemented based on FHS determination.

The determinations performed during the week before delivery showed for both groups a similar minimum (−5), maximum (+1), and median (−2) fetal head station. However, unlike the weekly evaluations, there were significant differences between the two vaginal and Cesarean delivery groups. Among the CS group, one-fifth presented with −3 and −1 FHS, and −4 and –2 stations were noted in one-quarter of cases. Most of the women that delivered vaginally were found with similar FHS (−4 to −2), but a different distribution with increasing and higher rates at more advanced stations: from 10% at station −4 to 35% at station −1.

To our best knowledge, the literature lacks studies on the delivery outcome based on the dynamic evaluation of the fetal head situation at term before the onset of labor. Therefore, we consider this research a good starting point for future larger studies. There is proof regarding the role of fetal head descent at the beginning of labor as a predictor for labor outcome [22]. In addition, in women undergoing labor induction at term, digital examination accurately predicted vaginal delivery within 24 h, while fetal fibronectin and ultrasound measurements failed to predict the outcome of induced labor [23] accurately. Our findings do not confirm the role of head engagement determination in standard term evaluations before labor onset [22,23,24,25]. In an attempt to predict the delivery mode by digital examination in early labor, several studies [22,24,25] found a considerably lower prediction rate for the Cesarean section delivery than in the study of Levy et al. [16], which used imagistic means to estimate fetal head situation before labor. The future will most likely provide objective ultrasound techniques that include fetal head position and descent evaluations at term, along with maternal history and clinical characteristics, to provide better predictions for labor outcomes. However, some institutions may have limited access to ultrasound or may not possess well-trained sonographers, such as in low income countries. In these circumstances, the clinical FHS determination might be essential to determine the delivery mode.

However, other clinical characteristics should not be neglected, as maternal weight, height, and gestational BMI are independent factors that are associated with an increased chance of caesarean section, and this has been demonstrated in numerous studies [25,26,27,28,29]. Although we investigated a low-risk group, we found significant differences between the vaginal and Cesarean groups in terms of age, weight, and BMI, and demonstrated that these parameters should be considered when predicting the delivery mode. Furthermore, we provided a multiple logistic regression equation for clinical practice that considers the predictive clinical variables at term: the fetal head situation, age, weight, height, and BMI. However, the fine tuning of this prediction should be achieved by increasing the studied population and number of centers involved.

## 6. Strengths and Limitations of This Study

### 6.1. Strengths

Our observational prospective cohort study of unselected low-risk primigravidae is the first study to assess, in a dynamic way, the clinical head station at term, previously proposed for labor outcome predictions. Our research serves as a beneficial framework where more data could be added for more accurate predictions.

### 6.2. Limitations

The concept of normalcy is population-based and our study enrolled only Caucasian women, a fact that may affect generalizability.

The study’s dataset is unbalanced, given the discrepancy between the rates and numbers of vaginal and Cesarean births. This fact is due to the unselected/consecutive enrollment of the pregnant primiparous women. Finally, more accurate data may be retrieved from ultrasound evaluations at term, because the intra-observer and inter-observer clinical variability of head station evaluations is significant, independently of the presence of caput succedaneum, significant molding, or head position.

## 7. Conclusions

In conclusion, we noticed that the clinical evaluation of fetal head station in primiparous women before the onset of labor has a limited potential to provide additional information regarding the prediction of the delivery mode. There is a potential benefit for the determinations performed within the week before delivery, which would require weekly assessments of the head station at term, a policy that is unlikely to be implemented. Another potential benefit was the labor outcome estimation in late-term or prolonged pregnancy. We require confirmation of these data in larger studies before counseling primiparous women at term before the labor onset based on the clinical fetal engagement data. New research should consider ultrasound determinations of the fetal head situation regarding the head station, occiput position, and maternal characteristics.

## Figures and Tables

**Figure 1 jcm-11-03274-f001:**
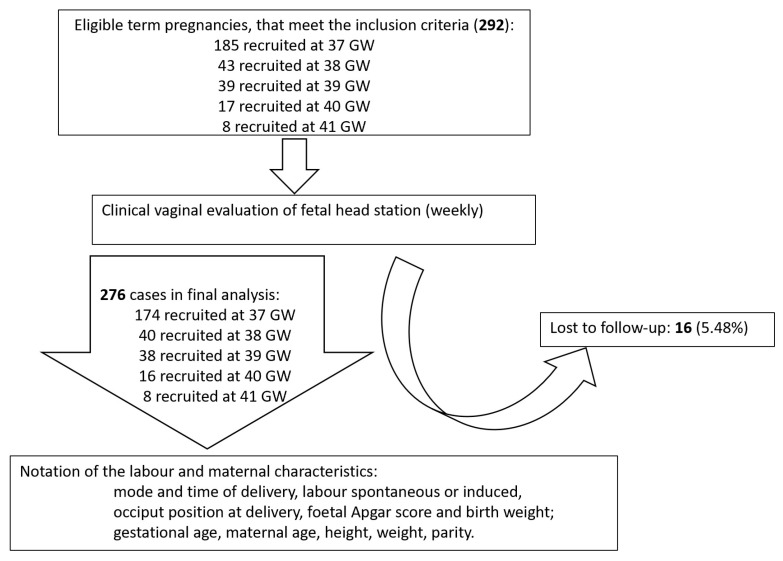
Flow chart summarizing the study group. GW, gestational weeks.

**Figure 2 jcm-11-03274-f002:**
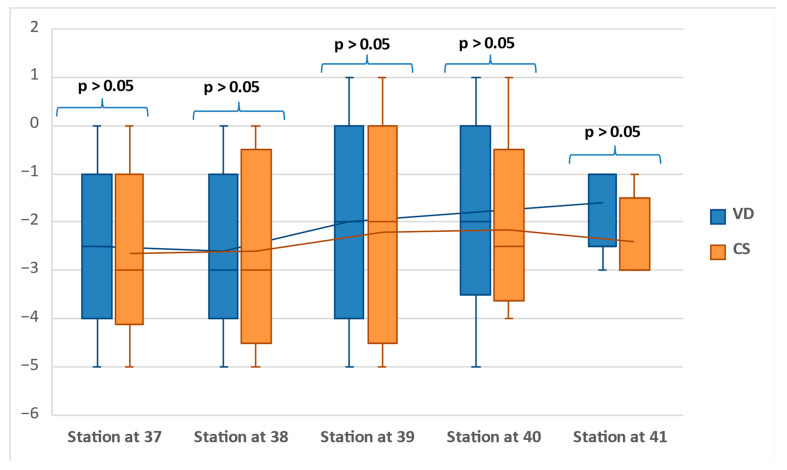
Box plot showing the distribution of the fetal head station at 37 to 41 weeks of gestation.

**Figure 3 jcm-11-03274-f003:**
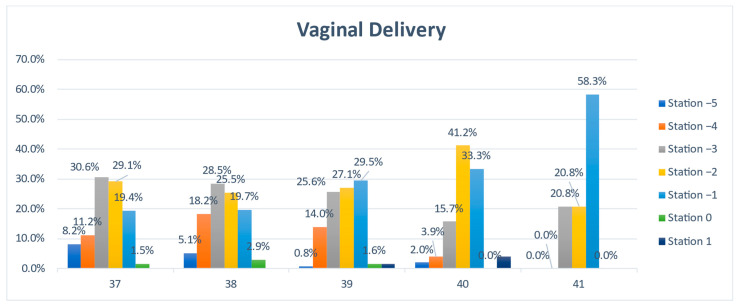
The comparative distribution of the clinical fetal head station at 37 to 41 weeks of gestation in the vaginal delivery group.

**Figure 4 jcm-11-03274-f004:**
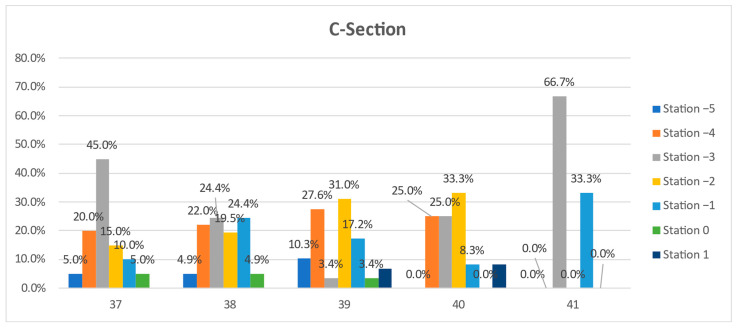
The comparative distribution of the clinical fetal head station at 37 to 41 weeks of gestation in the Cesarean delivery group.

**Figure 5 jcm-11-03274-f005:**
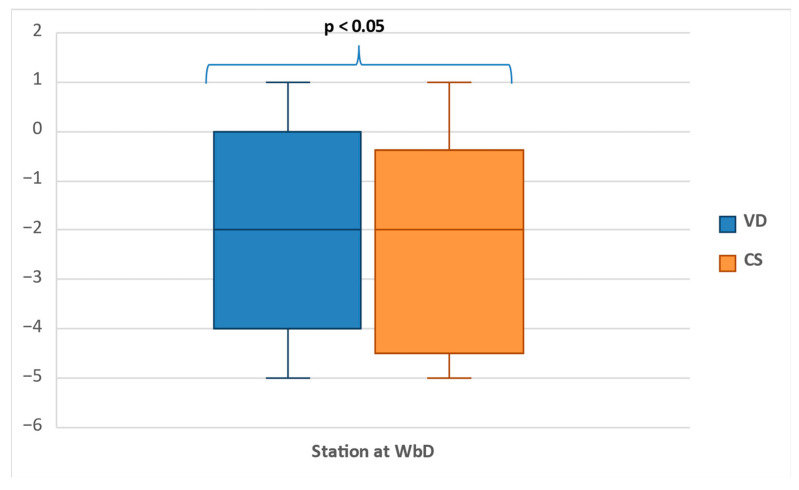
Box plot showing the distribution of the fetal head station in the week before delivery.

**Figure 6 jcm-11-03274-f006:**
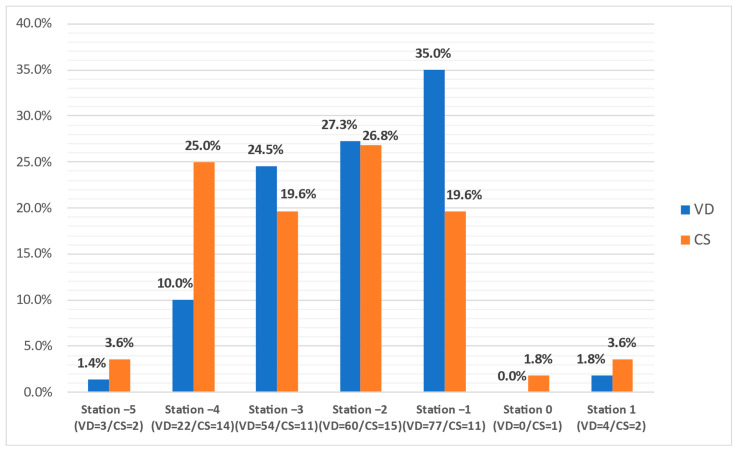
The comparative distribution of the clinical fetal head station in the week before delivery between the vaginal delivery group and the Caesarean section delivery group. VD, vaginal delivery; CS Caesarean section delivery.

**Figure 7 jcm-11-03274-f007:**
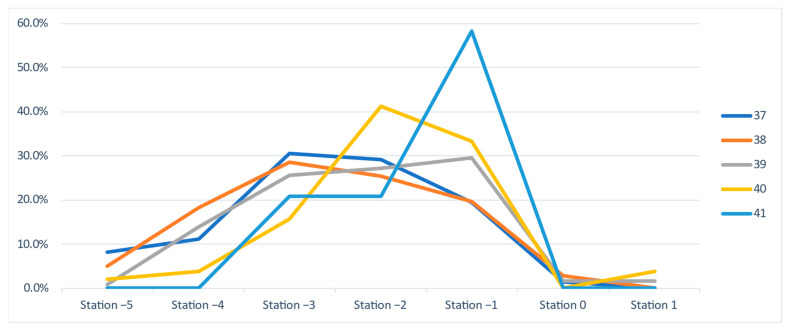
The rates of fetal head stations at each gestational week at term, in the group of vaginal deliveries.

**Figure 8 jcm-11-03274-f008:**
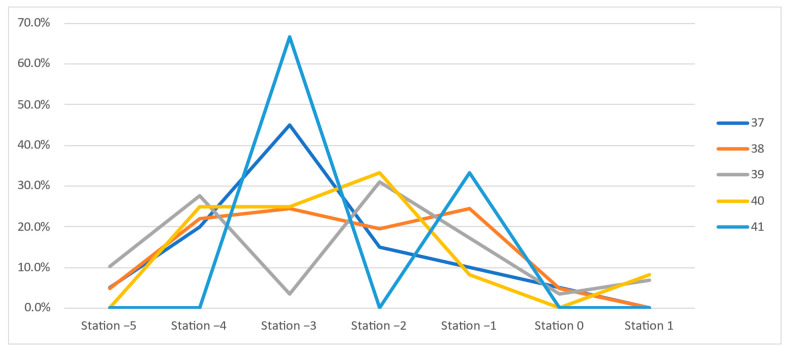
The rates of fetal head stations at each gestational week at term, in the group of Cesarean deliveries.

**Table 1 jcm-11-03274-t001:** Characteristics of study population of 276 nulliparous women at term and labor outcome.

Maternal and Labor Characteristics	Interval, Mean ± SD/Median
maternal age	18–35, 27.74 ± 3.845
maternal weight	52–109, 72.56 ± 9.121
maternal height	150–189, 167.21 ± 6.025
maternal BMI	20.07–38.37, 25.97 ± 3.062
maternal ethnicity	276 Caucasian
gestational age at inclusion	37w + 0d–41w + 2d
gestational age at delivery	37w + 1d–42w + 1d
mode of delivery	220 vaginal deliveries (79.7%) 56 Cesarean deliveries (20.3%)
Apgar score	5–10, 9
birth weight	2530–3960, 3395 ± 347.973
spontaneous and induced labor	256 spontaneous (92.8%), 20 induced (7.2%)
spontaneous vaginal delivery or instrumental	29 instrumental (13.2%% of vaginal births)

**Table 2 jcm-11-03274-t002:** *t*-test for independent variables results.

Variable	*t*-Value	*p*-Value
Age	−2.171	0.03
Weight	2.054	0.04
Height	−0.855	0.39
BMI	2.782	0.005

**Table 3 jcm-11-03274-t003:** Sums of square (SS), degrees of freedom (df), mean squares (MS), F-value, and *p*-level for overall goodness of fit through ANOVA.

Effect	SS	df	MS	F	*p*
	2.42	5	0.606	3.896	0.004

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
