# Peer review of "The Value of Fetal Head Station as a Delivery Mode Predictor in Primiparous Women at Term before the Onset of Labor"

_jcm, 2022, doi:10.3390/jcm11123274_

Round 1
Reviewer 1 Report
Dear authors,
thank you for sending the revised version of your manuscript to JCM.
I do have only three minor comments / questions:
Methods:
Please add: 4. „The clinical examinations were performed by experienced obstetricians (four senior physicians, with more than 5 years of practice).“
Results:
In fig. 1 you mean probably: „8 recruited at 41 GW“? - please correct.
Please provide the indications for Cesarean deliveries cases for better interpretation of your results. How many CDs were performed for obstructed labor? etc. Did you exclude emergency C-sections?
Thank you.
Author Response
#1. Dear authors, thank you for sending the revised version of your manuscript to JCM. I do have only three minor comments / questions:
Reply #1. We thank the reviewer for his/her time invested to help us upgrade the presentation of our study
#2. Methods: Please add: 4. „The clinical examinations were performed by experienced obstetricians (four senior physicians, with more than 5 years of practice).“
Reply #2. We agree the observation. The respective sentence was modified, page 2, line 109
#3. Results: In fig. 1 you mean probably: „8 recruited at 41 GW“? - please correct.
Reply #3. Indeed, the observation is correct. We apologize for the mistake; it was corrected in the new version of the manuscript.
#4. Please provide the indications for Cesarean deliveries cases for better interpretation of your results. How many CDs were performed for obstructed labor? etc. Did you exclude emergency C-sections?
Reply #4. We excluded elective C-sections and high obstetrical risk pregnancy, as mentioned in Methods section. We also excluded from the analysis the cases with CS with other than prolonged or arrested labor indication (fetal distress without prolonged labor, placental abruption etc.).
In order to clarify the exclusion criteria, the following paragraphs were modified and upgraded in the Methods section: “We included consecutive low-risk unselected primiparous women admitted for routine 3rd trimester scan, with a gestation age of more than 37 gestational weeks (GW), based on the first trimester dating scan. We excluded cases with indications for elective Cesarean delivery, and high obstetrical risk e.g., non-cephalic presentation, multiple pregnancies, preeclampsia, diabetes, fetal growth restriction and macrosomia, teenage and elderly primiparous, and prior Cesarean delivery.” (page 2, lines 60, 64-66)
“The Cesarean delivery cases with other than prolonged or arrested labor indication (fetal distress without prolonged labor, placental abruption etc.) were also excluded from the final analysis.” (page 2, lines 77-79)

Reviewer 2 Report
The authors present a prospective study that aimed to demonstrate the role of clinical determination of fetal head station in predicting mode of birth in primiparous women.
In general terms, the manuscript is well written structured and easy to follow, being clear in its objectives and results.
However, there are some major issues that need to be addressed, as follows:
- The authors do not show whether there are significant differences between the characteristics of the participants (Table 1) included in the two groups (natural birth vs. cesarean section). I consider that some of these characteristics influence the mode of delivery and if there are significant differences they should be discussed as limitations of the study.
- As I understand it, all caesarean sections were performed during labor. Why didn't the authors also consider cesarean sections that were indicated by an failure to initiate labor? In addition, it would be appropriate for the authors to present the indications for caesarean sections in included participants.
- Are there any participants in whom induction of labour has failed? What happened to these patients ( have they been excluded or included in the cesarean group)?
- Many other factors can influence the mode of delivery. Have the authors considered performing a multivariate analysis to predict the mode of delivery? I believe that this analysis (e.g. logistic regression) or discussion of this issue at the limitations of the study would be necessary.
Author Response
#1. The authors present a prospective study that aimed to demonstrate the role of clinical determination of fetal head station in predicting mode of birth in primiparous women.
In general terms, the manuscript is well written structured and easy to follow, being clear in its objectives and results.
However, there are some major issues that need to be addressed, as follows:
Reply #1. We thank the reviewer for his/her time invested to improve various aspects of our research.
#2. The authors do not show whether there are significant differences between the characteristics of the participants (Table 1) included in the two groups (natural birth vs. cesarean section). I consider that some of these characteristics influence the mode of delivery and if there are significant differences they should be discussed as limitations of the study.
Reply #2. We agree the observation of the reviewer. We did not present an analysis of the differences between the characteristics (Table 1), because we excluded from the study the pregnancies associated with a high obstetrical risk. We detailed this aspect in Methods section: “We included consecutive low-risk unselected primiparous women admitted for routine 3rd trimester scan, with a gestation age of more than 37 gestational weeks (GW), based on the first trimester dating scan. We excluded cases with indications for elective Cesarean delivery, and high obstetrical risk e.g., non-cephalic presentation, multiple pregnancies, preeclampsia, diabetes, fetal growth restriction and macrosomia, teenage and eldery primiparous, and prior Cesarean delivery.” (page 2, lines 60, 64-66)
Indeed, the differences between the two groups are not significat.
#3. As I understand it, all caesarean sections were performed during labor. Why didn't the authors also consider cesarean sections that were indicated by an failure to initiate labor? In addition, it would be appropriate for the authors to present the indications for caesarean sections in included participants.
Reply #3. We thank the reviewer for these observations meant to better clarify the study group characteristics. We considered the cases with induced labor, no matter the induction outcome. According to Table 1, 20 cases were induced (7.2%). Hence, the cases with progression failure following induction were included in the analysis. We agree that this aspect should be clearly stated, and we added in Methods section (page 2, line 67): “We did not exclude the cases with induced labor, indifferently the induction outcome.”
#4. Are there any participants in whom induction of labour has failed? What happened to these patients (have they been excluded or included in the cesarean group)?
Reply #4. Yes, we included these patients in the analysis. We addressed this issue in the Reply #3.
#5. Many other factors can influence the mode of delivery. Have the authors considered performing a multivariate analysis to predict the mode of delivery? I believe that this analysis (e.g. logistic regression) or discussion of this issue at the limitations of the study would be necessary.
Reply #5. The general study is designed to perform a multivariate analysis that takes into account ultrasound determinations regarding fetal head situation, occiput position and cervical length, along the clinical estimations presented here. We acknowledged the potential value of the ultrasound determinations both in Introduction and Discussion sections. Ultrasound measurements were recorded in this group, but we decided to publish separately the combined analysis results.
However, we consider that reporting the results of clinical evaluations is important, in a journal of Clinical Medicine. Digital vaginal evaluation in labor and especially before labor onset still represent the standard in most of the centers (due to logistics, resources and medical protocol reasons). Thus, physicians should be aware of the findings to calibrate their expectations regarding the clinical assessment at term.
In the last paragraph of Discussion section (lines 224-226), it is already stated that “The future will most likely provide objective ultrasound techniques that include fetal head position and descent evaluations at term, along with maternal history and clinical characteristics, to provide better predictions for labor outcomes”.

Round 2
Reviewer 2 Report
Thank you to the authors for taking the time to revise this paper. However, unfortunately, they did not provide a positive solution to some of the major problems of this study.
1. We considered that some participant characteristics (shown in Table 1) may influence the mode of birth. We therefore recommended that a comparison be made between the groups of participants (those who delivered by caesarean section and those who delivered naturally). For example, maternal weight, BMI and gestational BMI are independent factors that are associated with an increased chance of caesarean section, and this has been demonstrated in numerous studies (e.g. https://doi.org/10.1038/srep37168 ; https://doi.org/10.1186/s12884-020-03527-1 ). The same can be stated for maternal height (e.g. doi: 10.1371/journal.pone.0198124 ; http://dx.doi.org/10.1136/bmjopen-2021-054285). Therefore, given that these factors have been shown to influence the mode of birth, we felt that a statistical comparison between the two groups was necessary. The authors did not do so. They reasoned that they excluded high-risk pregnancies, but, as I argued in the lines above, other factors also influence mode of delivery. However, the authors replied that there is no significant difference between the two groups, but I think this needs to be proven in the paper.
2. We considered the need for multivariate analysis. By the objective of the study the authors say they want to demonstrate the role of clinical determination of fetal head status in predicting mode of birth. In order to be able to make a predictive analysis I consider it necessary to present odds ratios. This can be done by univariate and multivariate analysis. These analyses seem to be missing. I still consider it necessary to perform a multivariate analysis showing the odds ratio of natural/caesarean birth according to the stage of the fetal head, adjusted for factors that may influence the mode of birth (such as those presented above, e.g. weight, height, etc.).
Author Response
#1. Thank you to the authors for taking the time to revise this paper. However, unfortunately, they did not provide a positive solution to some of the major problems of this study.
Reply #1. We have to thank the reviewer, for his / her persistence led to a significant improvement of our manuscript.
#2. We considered that some participant characteristics (shown in Table 1) may influence the mode of birth. We therefore recommended that a comparison be made between the groups of participants (those who delivered by caesarean section and those who delivered naturally). For example, maternal weight, BMI and gestational BMI are independent factors that are associated with an increased chance of caesarean section, and this has been demonstrated in numerous studies (e.g. https://doi.org/10.1038/srep37168 ; https://doi.org/10.1186/s12884-020-03527-1 ). The same can be stated for maternal height (e.g. doi: 10.1371/journal.pone.0198124 ; http://dx.doi.org/10.1136/bmjopen-2021-054285). Therefore, given that these factors have been shown to influence the mode of birth, we felt that a statistical comparison between the two groups was necessary. The authors did not do so. They reasoned that they excluded high-risk pregnancies, but, as I argued in the lines above, other factors also influence mode of delivery. However, the authors replied that there is no significant difference between the two groups, but I think this needs to be proven in the paper.
Reply#2. The literature data were reviewed and found very useful. We also considered important to present and cite the previous contributions in Discussion section, lines 258-260:
“However, other clinical characteristics should not be neglected, as maternal weight, height and gestational BMI are independent factors that are associated with an increased chance of caesarean section, and this has been demonstrated in numerous studies (26-30). “
We performed a statistical comparison between the two groups regarding the maternal variables that have been shown to influence the delivery mode, and found significant differences between the two delivery modes in regard with the following features: mother’s age, weight, height, and BMI. What is more, we found that these parameters should be considered when predicting the delivery mode.
The following paragraphs were added to Results section (lines 189-209):
“The importance of maternal clinical characteristics at term
We were interested in seeing whether there are significant differences between the two delivery modes in regard with the following features: mother’s age, weight, height, and BMI. We have applied the t-test for independent variables. The results are shown in table 2.
Table 2. t-test for independent variables results
|
Variable |
t-value |
p-value |
|
Age |
-2.171 |
0.03 |
|
Weight |
2.054 |
0.04 |
|
Height |
-0.855 |
0.39 |
|
BMI |
2.782 |
0.005 |
From table 2, we can see that indeed there are significant differences (p-value < 0.05) between the two groups in terms of age, weight, and BMI.
Next, we were interested in performing a multiple logistic regression in which the predictive variable taken into account were the stage of fetal head, age, weight, height, and BMI. The obtained regression equation was:
We have also performed an overall goodness of fit through ANOVA. The obtained results in terms of sums of square (SS), degrees of freedom (df), mean squares (MS), F-value, and p-level are shown in Table 3.
Table 3. Sums of square (SS), degrees of freedom (df), mean squares (MS), F-value, and p-level for overall goodness of fit through ANOVA.
|
Effect |
SS |
df |
MS |
F |
p |
|
|
2.42 |
5 |
0.606 |
3.896 |
0.004 |
From table 3, we can see that indeed these parameters can be considered when predicting the delivery mode (p-level < 0.05).”
The findings were commented in Discussion section (lines 260-266):
“Although we investigated a low-risk group, we found significant differences between the vaginal and Cesarean groups in terms of age, weight, and BMI and demonstrated that these parameters should be considered when predicting the delivery mode. Furthermore, we provided for clinical practice a multiple logistic regression equation that takes into account the predictive clinical variables at term: the fetal head situation, age, weight, height, and BMI. Still, fine tuning of this prediction should be achieved by increasing the studied population and number of centers involved.”
#3. We considered the need for multivariate analysis. By the objective of the study the authors say they want to demonstrate the role of clinical determination of fetal head status in predicting mode of birth. In order to be able to make a predictive analysis I consider it necessary to present odds ratios. This can be done by univariate and multivariate analysis. These analyses seem to be missing. I still consider it necessary to perform a multivariate analysis showing the odds ratio of natural/caesarean birth according to the stage of the fetal head, adjusted for factors that may influence the mode of birth (such as those presented above, e.g. weight, height, etc.).
Reply#3. A multiple logistic regression was performed and added to the present manuscript, in which the predictive variable was taken into account. We presented above the results of this analysis and their integration in the paper.
The findings and their interpretation were also added in Abstract section, lines 22-30.
Round 3
Reviewer 2 Report
The authors have responded to almost all requests. I consider that at this moment the manuscript is suitable for publication in this form.
This manuscript is a resubmission of an earlier submission. The following is a list of the peer review reports and author responses from that submission.
Round 1
Reviewer 1 Report
Dear authors,
thank you for sending “The value of fetal head station as a delivery mode predictor in primiparous women at term before the onset of labor” to the Journal of Clinical Medicine.
The ability to predict the chance of a normal vaginal or operative delivery before the onset of labor would be of great usefulness in daily clinical practice to prevent fetomaternal labor complications while avoiding an increased rate of C-sections.
I refer here to your own comment published in the White Journal ten years ago (Ultrasound Obstet Gynecol. 2012 Sep;40(3):255-6).
So far literature does not prove possible to achieve this.
It is worth continuously trying to predict labor outcome before its onset as e.g. Dr. Youssef et al did in an interesting study about the impact of dynamic changes of fetal head station on labor outcome (J Matern Fetal Neonatal Med. 2021 Jun;34(12):1847-1854).
From my point of view this research has to be done at the present time using ultrasound techniques that include fetal head position and descent evaluations at term, along with maternal history and clinical characteristics, as you write correctly at the end of your manuscript. I miss this in your paper.
Nowadays you cannot evaluate head station any more by digital examination only. The intra-observer and inter-observer variability is too high - independently of the presence of caput succedaneum, significant molding, or head position - as shown in literature repeatedly.
It remains unclear how many „experienced obstetricians“ did the clinical examinations in your study cohort and what is your definition of „experienced“?
Why did not you use ultrasound as an additional mean in your study?
Let me add some critical aspects:
Introduction: This section does not convince the reader for your approach, if fetal head station assessment accuracy before labor is limited as you expound why do you opt for the clinical evaluation?
Methods: Description of statistical analysis is missing.
Results:
Page 2: I do not understand the paragraph of „Group characteristics“ at all. You write that you included only primiparous women and then „74% (N = 204) of the primiparous women were at their first pregnancy“? How many women were included at the study finally? 259? How many delivered vaginally, how many per C-section? A flow diagram would have been perhaps useful for the reader.
A table about patients’ characteristics (including neonatal data like birth weight etc.) is missing.
Page 3: Table 1 and figure 1 show more or less the same results. Usually non-significant results are not indicated in figures.
Page 6: “because of the low number of pregnancies” – you do not provide the exact numbers in figures, only percentages, why?
Page 8: It is irritating that there are so many mistakes throughout the manuscript. E.g. figure 9: Which one is the group of VDs? There are two figures 9A and two 9Bs?
I do not understand your explanations of the two figures 9: I do not see the “increasing rates of more favorable head stations” nor that station -4 is better represented in the CD group?
Discussion: where are your limitations of the study?
In this section there should be normally a discussion of the findings in the context of published literature.
References: I could not find in any database the text of "DrăguÈ™in Roxana-Cristina, Tudorache Ștefania, Iliescu Dominic Gabriel, Ultrasound longitudinal evaluation of fetal head en-gagement before the onset of labor. A clinical protocol and pilot study, ISI Proceedings, 2017.“ I am sorry.
Cited literature is partly outdated.
Yours sincerely,
Reviewer 2 Report
This study aimed to demonstrate the role of the fetal head station at term to predict delivery mode before labor. The result showed a limited value of FHS before delivery.
This study has some interesting points such as weekly based assessment of FHS in various ways and meaningful FHS after 41 weeks of gestation to determine delivery mode. However there seems to be major critical points to be discussed.
- Extensive English editing is needed.
- Methods: How many experienced obstetricians are involved in this study? FHS assessment is very subjective. Objective tools to examine FHS needs to be explained in this section. Is this study approved by IRB? IRB approval number should be written in this section.
- Results and statistical analysis : Statistical analysis needs to be in method section. The results do not need to describe every result as numbers which are in tables in this section. There are too many graphs and this needs to be cut down and summarized.
- Discussion: Discussion is not another section of results. Explanation and interpretation of results should be in this section. Limitation and strength are lacking.
- There are various objective methods especially ultrasound to determine FHS and subsequently delivery mode in primiparous women. Ultrasound is very useful and safely utilized in delivery room however some institution might have limited access to ultrasound or there are no well-trained sonographers such as in low income countries. In this circumstances, this FHS determination might be essential to determine delivery mode.